# Clinical Actionability of Genes in Gastrointestinal Tumors

**DOI:** 10.3390/genes16101130

**Published:** 2025-09-25

**Authors:** Nadia Saoudi Gonzalez, Giorgio Patelli, Giovanni Crisafulli

**Affiliations:** 1IFOM-ETS, The AIRC Institute of Molecular Oncology, 20139 Milan, Italy; nadia.saoudi@ifom.eu (N.S.G.); giorgio.patelli@ifom.eu (G.P.); 2Vall d’Hebron Institute of Oncology (VHIO), 08035 Barcelona, Spain; 3Niguarda Cancer Center, Department of Hematology, Oncology and Molecular Medicine, Grande Ospedale Metropolitano Niguarda, 20162 Milan, Italy; 4Department of Oncology and Hemato-Oncology, Università degli Studi di Milano (La Statale), 20122 Milan, Italy

**Keywords:** gastrointestinal (GI) tumors, actionable gene alterations, targeted therapies, precision oncology, predictive biomarkers

## Abstract

Precision oncology is witnessing an increasing number of molecular targets fueled by the continuous improvement of cancer genomics and drug development. Tumor genomic profiling is nowadays (August 2025) part of routine cancer patient care, guiding therapeutic decisions day by day. Nevertheless, implementing and distilling the increasing number of potential gene targets and possible precision drugs into therapeutically relevant actions is a challenge. The availability of prescreening programs for clinical trials has expanded the description of the genomic landscape of gastrointestinal tumors. The selection of the genomic test to use in each clinical situation, the correct interpretation of the results, and ensuring clinically meaningful implications in the context of diverse geographical drug accessibility, economic cost, and access to clinical trials are daily challenges of personalized medicine. In this context, well-established negative predictive biomarkers, such as extended *RAS* extended mutations for anti-EGFR therapy in colorectal cancer, and positive predictive biomarkers, such as MSI status, *BRAF* p.V600E hotspot mutation, *ERBB2* amplification, or even *NTRK1*, *NTRK2*, *NTRK3*, *RET*, and *NRG1* fusions across gastrointestinal cancers, are mandatory to provide tailored clinical care, improve patient selection for treatment and clinical trials, maximize therapeutic benefit, and minimize unnecessary toxicity. In this review, we provide an updated overview of actionable genomic alterations in GI cancers and discuss their implications for clinical decision making.

## 1. Introduction

Cancer is not merely a disease of organs but of genomes. This fundamental truth has shifted the oncology paradigm from a focus on anatomical tumor origin to one increasingly defined by molecular identity. In this context, the concept of “actionability” represents the ability to match specific genomic alterations with effective targeted therapeutic interventions and it has emerged as the cornerstone of personalized oncology [1]. Gastrointestinal (GI) tumors are one of the many areas of drug development for precision medicine and, being highly heterogeneous both in their molecular landscape and in their anatomical and histopathological origins, are a key example of the growing complexity of molecular oncology.

GI malignancies encompass a broad and diverse group of tumors arising from the digestive tract, including both luminal tumors and malignancies of solid organs like the pancreas and hepatobiliary system. They represent a substantial portion of the global cancer burden, with incidence and mortality accounting for over one-quarter and one-third of all cancer cases, respectively [2,3]. Despite therapeutic advances, many GI tumors remain incurable, especially when diagnosed or progressing to the metastatic stage, with five-year survival rates below 20% across several tumor types [2,3]. Conventional cytotoxic chemotherapy remains the backbone of treatment, but resistance is common, often driven by tumor-intrinsic genetic and epigenetic mechanisms [4]. At the same time, the rising incidence of early-onset GI cancers (particularly among patients under 50) calls for a renewed urgency in applying genomic tools not only to treat but also to better understand tumor biology across age groups and populations [5,6,7].

The development of next-generation sequencing (NGS) technologies has accelerated the identification of oncogenic drivers and ushered in a new treatment paradigm that incorporates precision oncology. This marks a sharp departure from the 1990s, when molecular understanding of genomes (and tumor genomes) was still very limited; the Human Genome Project and subsequent advents of high-throughput NGS in the 2000s transformed oncology into a data-driven discipline [8,9,10,11].

Molecular alterations in driver genes such as *KRAS*, *BRAF*, *FGFR2*, *IDH1*, *HER2*, and *NTRK1, NTRK2*, and *NTRK3* (*NTRK1-3*) fusions can confer sensitivity to targeted therapies across diverse GI histologies. This, in the context of the approval of tissue-agnostic therapies beyond GI tumors, like pembrolizumab for microsatellite instability (MSI) and high tumor mutational burden (TMB-H) tumors and larotrectinib or entrectinib for *NTRK1-3* fusions, further contributes to redefining the traditional boundaries between tumor type and therapeutic strategy [12,13].

Beyond the implications of agnostic biomarkers and their targets in GI tumors, there remains a window for biomarkers that exhibit context-specific behavior, such as the role of Claudin 18.2 in gastric cancer or the unique *EGFR* dependency of hotspot mutations such as *p.*V600E in *BRAF* or *p.*G12C in *KRAS* in the setting of colorectal cancer (CRC). This underscores the importance of understanding mechanistic biology to improve outcomes for patients with cancer.

This review examines the clinical actionability of genomic alterations in GI tumors, classifying them from agnostic biomarkers to non-agnostic and focusing on the intersection of biological relevance, therapeutic opportunity, and clinical implications, with these biomarkers present in ESMO guidelines for each tumor, in the ESMO recommendation for the use of NGS for patients with advanced cancers, and in the ESMO Scale for Clinical Actionability of molecular Targets (ESCAT) framework [14,15].

## 2. Tumor-Agnostic Biomarkers and Therapeutic Implications in GI Oncology

Prior to the advent of the genomic revolution, cancer treatments have been guided by the tumor tissue of origin and its histological subtype, shaping clinical guidelines, trial designs, and drug approvals around these parameters. However, new discoveries in genomic technologies and the rise of precision medicine have revealed molecular alterations that appear across different tumor types, regardless of their tissue of origin. This concept was fundamental to lead to new clinical trial designs that group patients by shared molecular features rather than by primary tumor location and in this context a classic example is represented by basket trials [16].

In the successful setting of immunotherapy, the tumor-agnostic approach was reinforced demonstrating efficacy across multiple cancer types by harnessing the immune system regardless of the organ of origin [17]. These advances have been facilitated by broader access to NGS and by the growing understanding of the genomic landscape of cancer, alongside a rapidly expanding repertoire of targeted therapies. Moving beyond the era of site-specific cancer management, tissue-agnostic biomarkers have gained increasing prominence across various histologies, including GI cancers.

This section provides an overview of tumor-agnostic biomarkers and their clinical relevance in GI cancers. It is important to note that the percentage of a single genomic alteration may vary among different studies. Figure 1 aims to summarize the currently approved therapies targeting gene alterations and their implications in GI oncology.

### 2.1. MSI-High/dMMR (Microsatellite Instability–High/Deficient Mismatch Repair)

The mismatch repair (MMR) system is composed of proteins that correct errors in the DNA strands. These proteins form complexes that recognize and repair mismatches, insertions, and deletions arising during DNA replication or caused by external DNA damage [18]. A deficiency in one or more of these proteins leads to a malfunction of the MMR system, significantly increasing the mutation rate and causing multiple errors in microsatellite regions. These regions are rich in tandem nucleotide repeats and are therefore particularly prone to replication errors. This process varies among tumor types and results in MSI [19,20].

MSI status, which is assessed by PCR, unlike MMR status, which is typically evaluated by immunohistochemistry (IHC), can be classified as microsatellite stability (MSS) or MSI when one or more than one microsatellite is affected [21,22]. To determine microsatellite stability by PCR, the most commonly used routine test analyzes five mononucleotide loci: BAT-25, BAT-26, NR-21, NR-24, and MONO-27, and different kits may be used in the research setting, considering ten or more regions [23].

Deficiency of MMR proteins can be either germline or sporadic. Most dMMR/MSI CRCs result from somatic alterations in one of the MMR genes. This phenotype is observed in approximately 15–20% of early-stage CRCs and in 3.5–6.5% of metastatic cases [24]. dMMR/MSI is also found in a significant proportion of other GI tumors, especially gastric adenocarcinomas (~19%) [25]. In contrast, the frequency is lower (<2%) in esophageal, biliary tract, and pancreatic cancers [26]. Germline deficiencies are typically associated with Lynch syndrome [27]. Sporadic cases are often related to somatic events that lead to hypermethylation of the *MLH1* promoter, frequently driven by mutations such as p.V600E *BRAF* in CRC [28,29].

dMMR tumors often exhibit high (>10 mutations per megabase) TMB (we address the definition and the relevance of this biomarker below), which leads to the generation of neoantigens that trigger immune responses but also upregulate immune checkpoints like PD-1/PD-L1. These three effects of dMMR constitute predictive biomarkers of response to immune checkpoint inhibitors (ICIs) [30,31]. These tumors are enriched in tumor-infiltrating lymphocytes, memory T cells, and cytotoxic T lymphocytes, making MSI the key biomarker for immunotherapy response in advanced cancers. Its identification across GI cancers can profoundly alter therapeutic strategies, often justifying ICI as a preferred approach, with potential for chemo-free and even organ-preserving treatment paths in the localized setting [32,33,34]. Routine testing of microsatellite (or MMR) status is now universally recommended in GI cancers, both in the localized and metastatic disease, due to its significant therapeutic implications.

An anti-PD-1 antibody was the first tumor-agnostic therapy approved by regulatory agencies FDA and EMA in 2017 for solid tumors exhibiting an MSI phenotype [35]. The phase II KEYNOTE-016 trial was the first basket study to evaluate the efficacy of anti-PD-1 antibody (pembrolizumab) in previously treated patients with metastatic cancer. It reported a 53% overall response rate (ORR) with 21% of patients with complete response (CR) among patients with a dMMR/MSI phenotype [36]. Responses were seen among patients with various GI malignancies, including metastatic CRC (mCRC), cholangiocarcinoma, and gastroesophageal, pancreatic, and small bowel cancers. Five pivotal trials (KEYNOTE-012, 016, 028, 158, and 164) evaluated the clinical activity of the same drug as an agnostic treatment for MSI advanced solid tumors [36,37,38,39,40]. To better summarize the concept, Table 1 summarizes the main activity found including GI cancer patients. Notably, patients with mCRC were enrolled in two trials: KEYNOTE-016, a Johns Hopkins University-sponsored trial; and KEYNOTE-164, a Merck-initiated trial that enrolled patients at 21 clinical sites across nine countries, testing pembrolizumab in patients with dMMR/MSI mCRC who were refractory to multiple prior lines of chemotherapy. In this case authors reported an ORR of 33%, with 95% of responders maintaining their response for at least 12 months [36]. The KEYNOTE-158 included 233 patients across 27 different tumor types, including gastric, cholangiocarcinoma, pancreatic, and small bowel cancers, but excluding mCRC, and reported an ORR of 34% with pembrolizumab, with the median duration of response not yet reached [38].

In addition, dostarlimab (an anti-PD-1 antibody) has also been approved for use with a tumor-agnostic indication for any dMMR/MSI tumor following progression. This is based on the GARNET phase I study which assessed the safety and efficacy of dostarlimab as monotherapy in endometrial and non-endometrial dMMR (or POLE-mutant) solid tumors [41]. This study included patients who had progressed on prior therapy, and most were cancers of GI origin. The ORR in dMMR tumors was 38.7% overall and 36.2% in patients with mCRC [41].

#### 2.1.1. Therapeutic Implications of MSI in CRC

As introduced, dMMR/MSI is identified in approximately 15–20% of localized CRC and 3.5–6.5% of metastatic cases [42,43,44]. It is consistently associated with a favorable prognosis in stage II CRC; however, its prognostic value in stage III remains controversial [45,46]. From a phenotypic point of view, dMMR/MSI CRC patients typically present with tumors located in the proximal colon, with a higher prevalence of mucinous histology, older age, and female sex [47].

In the pivotal KEYNOTE-177 phase III trial, previously untreated dMMR/MSI mCRC patients were randomized to receive pembrolizumab or standard therapy (chemotherapy plus targeted agents, according to investigators) [48,49,50]. Pembrolizumab exhibited a significant improvement in the primary endpoint of PFS and improving quality of life compared to chemotherapy. Although no significant difference in OS was observed in the final analysis, this may be due to the high percentage of crossover in the chemotherapy arm (60% of progressing patients later received an ICI) [51]. The recent CheckMate 8 HW phase III trial further advanced immunotherapy in this setting by showing that untreated dMMR/MSI mCRC patients presented significantly longer PFS when treated with nivolumab plus ipilimumab rather than with chemotherapy (72% vs. 15% at 24 months of follow up), along with fewer side effects [52]. Moreover, nivolumab plus ipilimumab showed superior PFS compared to nivolumab monotherapy (not reached vs. 39.3 months) [53]. In the second-line setting, the combination of PD-1 and CTLA-4 inhibitors (nivolumab and ipilimumab) is also approved for dMMR/MSI mCRC, based on the results of the phase II CheckMate 142 study [54].

Given the marked sensitivity of MSI mCRC to immunotherapy, the non-metastatic treatment landscape has been importantly tackled as well in recent clinical trials, although, differently from the metastatic setting, definitive data on clinical utility (and effectiveness) are still warranted [55]. In rectal cancer, a landmark study with neoadjuvant dostarlimab in patients with dMMR/MSI tumors showed a 100% clinical complete response (cCR) rate in all the patients included, enabling a non-operative, organ-preserving approach [33]. These results have generated strong interest in expanding the role of neoadjuvant ICIs in rectal cancer. Dostarlimab has since then been tested in MSI rectal and non-rectal tumors in the neoadjuvant setting, without surgery, in patients achieving a cCR [34]. In a cohort of 117 patients, all those patients with a PCR also showed clearance of circulating tumor DNA (ctDNA), suggesting that ctDNA could serve as a surrogate biomarker of treatment response (although ctDNA test sensitivity would benefit from further improvement to avoid false negatives in non-PCR cases). Further data on a solo ICI as a definitive treatment in non-metastatic rectal cancer followed by watch-and-wait approaches are emerging, confirming these pioneering findings. Neoadjuvant immunotherapy in dMMR/MSI colon cancer has shown remarkable efficacy. Both NICHE-2 (nivolumab + ipilimumab) and NICHE-3 (nivolumab + relatlimab) reported ≥97% pathological response rates and ≥68% complete PCR, along with excellent short-term recurrence-free survival [55,56]. These results of impressive anti-tumor activity underscore the importance of systematic molecular screening even in localized settings. Nevertheless, randomized evidence is needed to support the full integration of neoadjuvant ICIs into standard care for early-stage dMMR/MSI colon cancer [57]. Recent data presented at ASCO 2025 from the ATOMIC trial showed that adding atezolizumab, an anti-PD-L1 antibody, to adjuvant FOLFOX chemotherapy in patients with stage III MSI CRC led to a 10% improvement in 3-year PFS [58]. However, the trial did not include an atezolizumab monotherapy arm, raising questions about whether chemotherapy is necessary in the adjuvant setting, which is currently being addressed in an ongoing clinical trial (NCT05231850).

#### 2.1.2. Therapeutic Implications of MSI in Other GI Tumors

Approximately 5–20% of gastric adenocarcinomas and about 1% of esophageal or gastroesophageal junction (GEJ) carcinomas exhibit dMMR/MSI [26]. The percentage differences in MSI of gastric adenocarcinoma depends on geographical factors, tumor stage, and age. Retrospective studies suggest that dMMR/MSI gastric cancers tend to occur in younger patients, are more frequently poorly differentiated, and are often associated with the diffuse histological subtype, predominantly affecting the antrum [59]. Gastric cancer can be divided in four molecular subtypes (based on TCGA molecular classification): Epstein–Barr virus (EBV) tumors (characterized by *PIK3CA* mutations, PD-L1/2 and JAK2 amplifications, and extensive CpG island methylation); MSI tumors with hypermutation and MLH1 silencing; genomically stable tumors enriched in diffuse histology and carrying RHOA mutations or CLDN18–ARHGAP fusions; and chromosomal instability (CIN) tumors (characterized by aneuploidy, frequent *TP53* mutations, and amplifications of receptor tyrosine kinases such as *ERBB2*, *EGFR*, and *MET*) [60]. In addition, recent studies have underscored the importance of gene regulatory networks and master regulators in distinguishing intestinal versus diffuse histotypes and in defining location-specific transcriptional programs with prognostic and therapeutic relevance, further supporting the need for stratified genomic approaches that go beyond single biomarker testing [61,62].

In the metastatic setting, MSI status has consistently emerged as a predictive biomarker of response to immunotherapy, based on exploratory biomarker analyses from phase III trials. In the second-line setting, a retrospective analysis of KEYNOTE-061 showed improved survival with pembrolizumab in MSI-H tumors, regardless of PD-L1 status [63]. In first-line treatment, exploratory analyses from KEYNOTE-062 and CheckMate 649 similarly demonstrated enhanced benefit from immune checkpoint inhibitors in MSI subgroups, independent of treatment backbone [64,65]. As a result, ESMO guidelines now recommend MSI testing for all patients with advanced gastroesophageal cancers.

Additionally, the benefit of perioperative or adjuvant chemotherapy in patients with MSI localized tumors is questionable. A post hoc analysis of the MAGIC trial indicated that dMMR tumors derived limited (or even detrimental) benefit from chemotherapy, whereas patients who underwent surgery alone had better outcomes [66]. These findings underscore the need for alternative strategies in the management of localized dMMR/MSI gastroesophageal adenocarcinomas. Given the robust activity of ICIs in metastatic MSI tumors, recent trials have explored their integration in the curative setting. The NEONIPIGA trial (phase II) evaluated neoadjuvant nivolumab plus ipilimumab followed by surgery in MSI gastric and GEJ cancers, reporting a PCR in 59% of patients and no progression under neoadjuvant therapy, suggesting a strong rationale for immunotherapy-based perioperative regimens [67]. Similarly, the INFINITY trial (also phase II) investigated nivolumab monotherapy or combined with ipilimumab in the neoadjuvant setting for dMMR/MSI tumors and included a non-operative management (NOM) arm for patients achieving a cCR, supporting the feasibility of organ preservation in highly selected cases [68]. In contrast, the MATTERHORN trial incorporated durvalumab into perioperative FLOT chemotherapy in resectable, HER2-negative gastric and GEJ adenocarcinoma but enrolled an unselected, all-comer population regardless of MSI status. While it reflects the growing interest in immunotherapy in localized disease, MATTERHORN does not specifically inform the management of MSI-H tumors, which may require chemotherapy-free approaches. Collectively, these trials challenge the traditional reliance on perioperative chemotherapy and open the door to biomarker-driven strategies, including NOM, for patients with MSI gastroesophageal cancers.

In biliary tract cancers, dMMR/MSI is found in up to 10% of intrahepatic and ampullary tumors and is associated with Lynch syndrome. Early trials like KEYNOTE-158 demonstrated durable responses to pembrolizumab in this subgroup, leading to its approval for previously treated dMMR/MSI cases, though further data on optimal combinations and timing are needed [38].

MSI tumors, though rare in pancreatic cancer (1–2%), represent a relevant targetable subgroup. The first study showing the potential of PD-1 blockade across MSI tumors included 22 pancreatic cancer patients, with limited responses (1 CR = 4%; 3 PR = 14%) and a median OS of 4 months [36]. Nevertheless, subsequent retrospective cohort studies have indicated that careful patient selection can improve immunotherapy outcomes, with some series reporting an ORR as high as 48% [69,70,71].

Small bowel adenocarcinomas, despite being a rare disease with limited data, display a relatively high rate of dMMR/MSI (10–20%) and responses to ICIs such as pembrolizumab and dostarlimab, supporting their use in this population [38,41].

Conversely, in hepatocellular carcinoma, dMMR/MSI tumors are exceedingly rare, accounting for less than 1% of cases [26]. Although data specific to this subgroup remain limited, isolated clinical responses to ICIs, such as dostarlimab, have been documented [41]. Despite this scarcity of MSI-specific evidence, ICIs (particularly combinations such as atezolizumab plus bevacizumab and durvalumab plus tremelimumab) are now established as the first-line standard of care (SOC) for advanced HCC, independent of the tumor’s MSI/MMR status [72].

### 2.2. TMB-High (Tumor Mutational Burden ≥ 10 Mut/Mb)

TMB, defined as the number of (all or only non-synonymous) somatic mutations per megabase (mut/Mb), has emerged as a potential predictive biomarker for response to ICIs across multiple tumor types [73,74]. TMB-H is thought to increase neoantigen load, enhancing T cell activation and response to therapies targeting PD-(L)1 and CTLA-4. While whole-exome sequencing (WES) is the gold standard for TMB assessment, large enough targeted panels are more widely used due to lower complexity and costs, though variability exists among platforms [75,76,77].

Although GI cancers generally have lower TMB than melanoma or lung cancer, TMB-H is still observed in certain subtypes. For example, it has been identified in 14.6% of right-sided colon cancers, 10.2% of small bowel adenocarcinomas, and 8.3% of both anal and gastric cancers; it is less frequent in pancreatic (1.4%) and esophageal adenocarcinomas (1.9%) [78,79,80]. TMB-H is strongly enriched in MSI cases for most GI tumors, except anal and some gastric cancers [79]. Additionally, *POLE*/*POLD1* mutations are known to associate with TMB-H [81,82]. While many TMB-H GI cancers already qualify for ICIs via MSI status, the tumor-agnostic approval of pembrolizumab extends therapeutic options to MSS TMB-H tumors such as esophageal, anal, and a few CRCs [83]. Despite its promise, challenges remain in optimizing TMB thresholds and confirming efficacy across MSS GI subtypes.

In the phase II KEYNOTE-158 trial, 790 patients across nine different tumor types (excluding mCRC) were treated with pembrolizumab and stratified based on their TMB status. A retrospective analysis of the KEYNOTE-158 trial showed that patients with TMB-H (≥10 mut/Mb) had an ORR of 29% to pembrolizumab, compared to 6% in non-TMB-H tumors, regardless of MSI status [84]. These findings led to the FDA’s 2020 approval of pembrolizumab for unresectable or metastatic TMB-H tumors that progressed after prior treatment. However, response among TMB-H GI cancers, such as anal cancer, was modest (e.g., only one responder among fourteen TMB-H cases in KEYNOTE-158 representing about 7%). Additionally, a CRC-focused study revealed that a subset of tumors with TMB-H but lacking both MSI-H and *POLE*/*POLD1* mutations exists, representing a genomically undefined TMB-H subgroup but demonstrating limited clinical benefit from ICIs. Despite meeting the FDA-approved TMB ≥10 mut/Mb threshold, these tumors showed a poor response rate and short-living PFS compared to MSI-H or *POLE*-mutated counterparts. This finding highlights how using TMB as a pan-tumor biomarker is still debated, especially in CRC, and encourages the importance of contextualizing TMB within the molecular and immunologic landscape of each tumor type [85,86].

Although the cutoff of ≥10 mut/Mb is FDA-approved, several studies have adopted higher thresholds, such 13, 16, or even 20 mut/Mb to better enrich for tumors with a truly high neoantigen burden in different settings [86,87,88]. Moreover, additional variability arises from the conflation between blood-derived TMB (bTMB) and tissue-based TMB (TMB), which differ significantly in terms of technical characteristics and clinically meaningful cutoffs. This distinction becomes particularly relevant in advanced disease settings, where ctDNA release is influenced by the number, size, and anatomical distribution of metastatic lesions [89,90].

### 2.3. BRAF p.V600E Mutation

The mutation in the p.V600E hotspot of the *BRAF* gene plays a central role in tumorigenesis through aberrant activation of the MAPK signaling pathway. Initially identified in melanoma, *BRAF* mutations have since been detected in multiple solid tumors, including CRC, non-small-cell lung cancer (NSCLC), thyroid carcinoma, biliary tract cancers (BTCs), and gastrointestinal stromal tumors (GISTs), as well as in certain hematologic malignancies and pediatric tumors. This widespread occurrence underscores the fundamental role of *BRAF* downstream signaling in cell proliferation and survival, establishing it as a cornerstone biomarker in precision oncology [12].

From a molecular standpoint, *BRAF* mutations can be classified according to their functional impact and effect on BRAF dimerization: class I mutations, typified by the V600E variant, are RAS-independent and activate BRAF as monomers, resulting in constitutive MEK–ERK pathway activation, while class II mutations are also RAS-independent but require dimerization to be constitutively active. In contrast, class III mutations are kinase-impaired and depend on upstream RAS activity for MAPK pathway activation [91]. Notably, the clinical significance of *BRAF* mutations transcends tumor type. This realization led to the VE-BASKET trial, one of the first tumor-agnostic studies to enroll patients based on the presence of *BRAF* V600E mutations regardless of histology [92]. The trial demonstrated the efficacy of vemurafenib across various *BRAF*-mutated non-melanoma cancers, providing proof of concept for histology-independent, molecularly guided therapies. Following disappointing outcomes of BRAF inhibitor monotherapy in CRC, the trial protocol was amended to include the addition of EGFR inhibitors (specifically, vemurafenib combined with cetuximab) in the CRC cohort [93,94].

Since then, evidence rapidly evolved, and the FDA finally granted accelerated approval for the combination of dabrafenib (anti-BRAF) and trametinib (anti-MEK) for adult and pediatric patients (aged ≥ 6 years) with unresectable or metastatic solid tumors harboring a *BRAF* V600E mutation. Among the patients enrolled in the BRF117019 and NCI-MATCH trials used for the approval, several GI tumor types were included, such as cholangiocarcinoma, small bowel adenocarcinoma, pancreatic adenocarcinoma, neuroendocrine tumors, mucinous–papillary serous adenocarcinoma of the peritoneum, and anal adenocarcinoma [95,96].

In CRC, *BRAF* mutations (mostly p.V600E) occur in approximately 8–15% of cases and are almost always mutually exclusive with *KRAS* mutations. *BRAF*-mutant CRC exhibits a distinct clinical phenotype, being more frequent in older female patients and typically located in the proximal colon. Histologically, these tumors often show poor differentiation, mucinous features, larger primary size, and a greater tendency for nodal and peritoneal dissemination. Importantly, *BRAF* p.V600E mutations are associated with MSI status, due to their role in the CpG island methylator phenotype (CIMP), which leads to epigenetic silencing of the *MLH1* gene. Approximately 30% of *BRAF*-mutant CRCs are MSI. Clinically, *BRAF* p.V600E mutations are associated with poor outcomes, particularly in MSS tumors, with a median survival of ~13.4 months versus ~37 months in wild-type tumors and a 5-year survival of 47.5% versus 60.7%, respectively [97].

The identification of *BRAF* mutations has paved the way for targeted treatment strategies in CRC. However, as previously highlighted, early trials using BRAF inhibitors as monotherapy in *BRAF* p.V600E-mutant metastatic mCRC yielded limited clinical benefit, primarily due to rapid feedback reactivation of EGFR and sustained MAPK signaling, mechanisms that are absent in melanoma. Preclinical studies revealed that BRAF inhibition alone led to EGFR upregulation, providing a strong rationale for combining BRAF and EGFR inhibitors in mCRC [93,94]. These insights culminated in the BEACON CRC trial, a pivotal phase III study demonstrating that dual therapy with encorafenib (a selective BRAF inhibitor) and cetuximab (an EGFR-targeting antibody) significantly improved OS (8.4 vs. 5.4 months) compared with standard chemotherapy in chemotherapy-pretreated patients (>1 line of treatment). The addition of the MEK inhibitor binimetinib yielded a slight increase in survival (9.0 months) but also higher toxicity [98,99]. Based on these findings, encorafenib plus cetuximab was approved in 2020 for second-line treatment of *BRAF* V600E-mutant mCRC.

More recently, the BREAKWATER trial has explored the use of encorafenib and cetuximab in combination with chemotherapy in the first-line setting for *BRAF* V600E-mutant mCRC. Preliminary results showed a median OS of 30.3 months for the triplet combination, compared to 15.1 months with mFOLFOX alone, establishing a new standard of care that repositions the prognosis of BRAF p.V600E-mutant mCRC in line with other molecular subtypes [100]. Ongoing trials like SEAMARK (NCT05217446) are further evaluating the addition of immune checkpoint inhibitors in MSI-H populations.

In BTC, *BRAF* mutations are found in 5–7% of cases, predominantly in intrahepatic cholangiocarcinoma. *BRAF* p.V600E in this context is associated with more advanced disease at diagnosis, resistance to standard chemotherapy, and reduced OS (13.5 vs. 37.3 months) [26]. Despite their low frequency, BRAF-targeted therapies have shown promise: the ROAR trial reported a 53% ORR and 9-month median PFS with dabrafenib plus trametinib [101]. Likewise, in the NCI MATCH EAY131-H trial, 75% of patients with *BRAF* V600E-mutated BTC experienced PR [96].

Though less common, *BRAF* p.V600E mutations also occur in a small subset of GISTs (0.6–3.9%). In these cases, *BRAF* alterations function as tumor-agnostic predictive biomarkers, similar to *NTRK1*, *NTRK2*, and *NTRK3* (*NTRK1-3*) gene fusions, supporting the rationale for broad molecular profiling and precision targeting strategies in rare GI tumors [12].

### 2.4. HER2 Overexpression/ERBB2 Amplification

HER2 (*ERBB2* is the name of the gene encoding the protein) is a transmembrane receptor tyrosine kinase (TK) whose amplification or overexpression drives tumorigenesis through MAPK and PI3K signaling. While originally targeted in breast cancer, HER2 has emerged as a critical biomarker in several GI tumors, including gastric, colorectal, biliary tract, and pancreatic cancers. On 5 April 2024, the FDA granted accelerated, tumor-agnostic approval to trastuzumab deruxtecan (Enhertu, T-DXd) for adults with unresectable or metastatic HER2-positive (IHC 3+) solid tumors who had received prior systemic therapy and lacked satisfactory treatment alternatives [102]. This decision was based on robust response rates and duration of response observed across three phase II trials, DESTINY-PanTumor02, DESTINY-Lung01, and DESTINY-CRC02, confirming the broad anti-tumor activity of this HER2-targeted antibody–drug conjugate (ADC) across diverse cancer types, including gastrointestinal malignancies such as colorectal and biliary tract cancers [103,104,105].

HER2 overexpression or gene amplification is typically assessed through a combination of IHC and fluorescence in situ hybridization (FISH). A HER2 IHC score of 3+, defined by strong complete membranous staining in ≥10% of tumor cells, is considered positive. Cases with a 2+ IHC score (equivocal) require confirmatory FISH analysis to determine gene amplification [106]. While HER2-targeted therapies have revolutionized the management of advanced HER2-positive breast cancer since the late 1990s with approved agents including trastuzumab, pertuzumab, T-DXd, trastuzumab emtansine (T-DM1), tucatinib, lapatinib, and margetuximab, the clinical relevance of HER2 extends beyond breast cancer.

In gastric and GEJ adenocarcinomas, HER2 positivity occurs in 10–20% of cases and is routinely assessed via IHC and FISH [107]. HER2-targeted therapy was first validated in the ToGA trial, where trastuzumab plus chemotherapy improved survival over chemotherapy alone [108]. For refractory disease, the DESTINY-Gastric01/02 trials confirmed that the ADC T-DXd yields high response rates (42–51%) and median OS of over 12 months [109,110]. Novel agents like zanidatamab and RC48-ADC are also showing promise, especially in trastuzumab-exposed patients [111]. In patients with gastric cancer, there is a high co-expression of PD-L1 and HER2 in 85% of patients. Building on this, the KEYNOTE-811 trial led to EMA and FDA approval of pembrolizumab + trastuzumab + chemotherapy in HER2+/PD-L1+ tumors, showing an improvement not only in PFS but also in OS [112].

In mCRC, *ERBB2* amplification is seen in ~3–4% of cases, particularly in *RAS/BRAF* wild-type tumors and more distal positions such as rectal cancers, and predicts resistance to anti-EGFR agents (from retrospective data) [113,114]. Several HER2-targeted strategies have shown promising activity in HER2-positive, *RAS* wild-type mCRC. The HERACLES trials explored combinations of trastuzumab with lapatinib or pertuzumab, showing clinical benefit in pretreated patients [115]. Later, other combinations were tested, mostly employing trastuzumab and pertuzumab [116,117,118]. T-DXd demonstrated encouraging results in heavily pretreated cases, including those previously exposed to anti-HER2 agents [86]. Based on these advances, the ongoing MOUNTAINEER-03 trial is evaluating first-line HER2 blockade combined with chemotherapy to potentially establish a new standard of care [119].

HER2 overexpression (IHC 3+) is uncommon in pancreatic cancer, and T-DXd showed its most modest ORR of 4% in this specific population in the DESTINY-PanTumor02 trial [105]. Nonetheless, some responses were observed upon independent review, suggesting further exploration may be warranted.

### 2.5. Fusions in NTRK1, NTRK2, and NTRK3 Genes

TRKA-C gene fusions, involving *NTRK1-3*, result in constitutively active TRK fusion proteins that promote oncogenic signaling, most notably through MAPK and PI3K pathways. Though rare in GI malignancies overall, these fusions represent critical therapeutic targets in select tumor subtypes [26].

In CRC, NTRK fusions are identified in approximately 0.2–0.3% of unselected cases, with enrichment in MSI, *RAS*/*BRAF* wild-type tumors, particularly those with a sessile serrated pathway origin. Clinically, these tumors may present with right-sided location and mucinous histology. Though infrequent, *NTRK1-3* fusions are relevant due to their strong predictive value for response to TRK inhibitors. A pan-cancer analysis including 14 patients with GI tumors treated with larotrectinib reported a 50% response rate in mCRC, with durable responses even in heavily pretreated patients. Similarly, entrectinib has shown meaningful activity in mCRC and other GI tumors with NTRK fusions, with an ORR of 43% in pooled GI cohorts [120].

In cholangiocarcinoma, *NTRK1-3* fusions are estimated to occur in about 3.6% of cases, particularly intrahepatic subtypes, and represent one of the few actionable alterations beyond *FGFR2* fusions and *IDH1* mutations [121,122]. Pancreatic ductal adenocarcinomas, by contrast, harbor *NTRK1-3* fusions far less frequently (~0.3%), though modest enrichment was reported in *RAS* wild-type cases and individual case reports document durable responses to TRK inhibitors [69,123,124].

Accurate identification of these fusions is essential but challenging. Although methods such as FISH and reverse transcriptase polymerase chain reaction (RT-PCR) are available, their limited sensitivity and need for prior knowledge of fusion partners reduce their utility in GI tumors. NGS, especially RNA-based platforms, offers higher sensitivity and the ability to detect novel or rare fusion events. IHC for pan-TRK proteins may serve as a cost-effective screening tool, though confirmatory NGS is advised in positive cases due to potential false positives, especially in GI tumors with physiological TRK expression [125].

Whole-genome sequencing (WGS) can also be employed to identify fusion events, with bioinformatic pipelines specifically designed to detect gene rearrangements and characterize their breakpoints [126]. In addition, there are targeted gene panels developed for specific clinical purposes, ranging from comprehensive genomic profiling to minimal residual disease (MRD) settings, that capture the sequences of *NTRK1-3* genes and canonical fusion partners, achieving good performance and reliable detection rates [89,127].

Given the promising efficacy of TRK inhibitors, particularly larotrectinib and entrectinib which are both FDA- and EMA-approved for tissue-agnostic indications and have favorable safety profiles, routine *NTRK1-3* fusion testing should be considered in the molecular workup of GI cancers. This is especially relevant for MSI mCRC, *RAS/BRAF*-wild-type GI tumors (particularly in pancreatic cancer), refractory or rare histologies, and cases lacking other actionable drivers. Also, in the case of pancreatic cancer, guidelines recommend the study of rare gene fusions such as *NTRK* and *RET*, which, although uncommon, may offer access to highly effective targeted therapies.

### 2.6. RET Fusions

*RET* fusions represent oncogenic alterations arising from chromosomal rearrangements that juxtapose the RET kinase domain with promoter region from various partner genes. This leads to constitutive activation of downstream signaling pathways involved in cell proliferation and survival. While RET plays an essential role in embryonic development, particularly in renal and neural tissue, its aberrant activation contributes to tumorigenesis in several cancers, notably papillary thyroid carcinoma and NSCLC [128].

Although *RET* fusions are rare in GI malignancies, typically occurring in <2% of cases, their detection carries significant therapeutic implications. The LIBRETTO-001 trial assessed the RET inhibitor selpercatinib in patients with RET-fusion-positive tumors beyond thyroid and lung cancers. Out of the 41 patients, 23 (56%) had GI, including cholangiocarcinoma, and CRC. The ORR in the GI cohort was 44%, with a median duration of response (DOR) of 18.4 months, indicating durable efficacy even in heavily pretreated cases [129]. Similarly, the RET inhibitor pralsetinib demonstrated a 53% ORR across *RET*-fusion-positive non-NSCLC/non-thyroid tumors, with particularly promising results in pancreatic cancer (3/3 responders) and cholangiocarcinoma (2/3 responders) [130].

From a molecular standpoint, *RET* fusions in GI cancers often involve diverse upstream partners, making broad-based genomic profiling, preferably via RNA-based NGS while less common via DNA-based workflow, essential for accurate detection. Specifically, as with *NTRK1-3* fusion detection, RNA sequencing remains the preferred approach, although DNA-based assays can be employed when using gene panels that capture RET and its canonical fusion partners or through WGS data. Immunohistochemistry and FISH are less commonly used in this context due to variability in expression and limited fusion partner specificity [128].

Despite their low prevalence, *RET* fusions have emerged as a clinically actionable biomarker. Based on durable responses across histologies, selpercatinib has received tumor-agnostic EMA and FDA approval, reinforcing the importance of including RET in comprehensive molecular diagnostic panels for GI malignancies—especially in pancreatic and biliary tract tumors where therapeutic options are limited [131].

## 3. Non-Agnostic Biomarkers and Therapeutic Implications in GI Oncology

While tumor-agnostic therapies have expanded treatment options for a subset of patients, many clinically relevant genomic alterations in GI cancers remain organ-specific in terms of approved targeted therapies. Several biomarkers, such as *BRCA1/2* mutations in pancreatic cancer, *IDH1* and *FGFR2* alterations in cholangiocarcinoma, Claudin 18.2 expression in gastric cancer, the hotspot *KRAS* p.G12C in CRC and pancreatic cancers, and *NRG1* fusions across multiple GI tumor types, have emerged as promising therapeutic targets. In this section, we review the most prevalent and emerging non-tumor-agnostic molecular alterations in GI malignancies, highlighting their biological rationale, prevalence, and evolving therapeutic landscape. We summarize the concept in Figure 2 illustrating the main non-agnostic biomarkers and their corresponding targeted therapies.

### 3.1. NRG1

The first biomarker is Neuregulin-1 (NRG1) that is a member of the EGF family involved in the development of the nervous and cardiovascular systems. It acts as a ligand for ERBB3 and ERBB4 receptors, promoting heterodimerization, especially with ERBB2, and triggering oncogenic signaling through the PI3K-AKT and MAPK pathways [132]. Gene fusions involving *NRG1* lead to aberrant surface expression of the EGF-like domain, resulting in constitutive ErbB2/ErbB3 activation and persistent downstream signaling that drives tumorigenesis [132]. 

Although rare (estimated in ~0.2% of all solid tumors) *NRG1* fusions have been identified in several GI malignancies, including pancreatic ductal adenocarcinoma (~0.5%) (particularly in *KRAS* wild-type tumors), cholangiocarcinoma (~0.5%), and CRC (~0.1%) [133,134,135]. Detection of these fusions can be challenging due to diverse fusion partners, involvement of untranslated regions, and large intronic sequences, and it is difficult to use custom panels. RNA-based sequencing has demonstrated superior sensitivity over DNA-based NGS for identifying these complex rearrangements [136]. Afatinib, a pan-ErbB TK inhibitor, showed tyrosine activity in *NRG1*-fusion-positive tumors, with partial responses reported in some GI cancer patients [134,136]. Recently, clinical data from zenocutuzumab, a first-in-class, humanized bispecific IgG1 antibody targeting HER2 and HER3, have been published. In a tumor-agnostic phase II clinical trial, 204 patients with advanced solid tumors harboring *NRG1* fusions were treated with zenocutuzumab. Among 158 evaluable patients, the ORR was 30%, with a median duration of response of 11.1 months and PFS of 6.8 months. Remarkably, responses were observed across multiple tumor types, including a 42% ORR in pancreatic adenocarcinoma, two PRs in ten cholangiocarcinoma patients, one PR among six CRC cases, and a PR in the only patient with gastric cancer [137].

### 3.2. Claudin 18.2

Claudin 18.2 is a tight junction protein physiologically restricted to gastric mucosa. Upon malignant transformation, loss of cell polarity exposes CLDN18.2 on the tumor cell surface, making it accessible to targeted therapies. CLDN18.2 is not expressed in most normal tissues and is not prognostic [138]. It is typically not co-expressed with PD-L1, tends to be enriched in patients with low PD-L1 CPS, and shows no clear association with HER2 status [139]. Claudin 18.2 was found expressed in approximately 38% of HER2-negative advanced gastric and GEJ adenocarcinomas. In this case authors used a moderate to strong staining cut-off (≥2+) showing ≥75% of tumor cells, which was the eligibility criterion for the pivotal phase III studies [140].

Zolbetuximab is a first-in-class monoclonal antibody that specifically targets CLDN18.2. It binds to CLDN18.2-expressing cancer cells. Two pivotal phase III trials, SPOTLIGHT and GLOW, have established the clinical benefit of adding zolbetuximab to first-line chemotherapy in patients with CLDN18.2-positive, HER2-negative advanced GI or GEJ adenocarcinoma [141,142]. The SPOTLIGHT trial evaluated zolbetuximab in combination with mFOLFOX6 and demonstrated a significant improvement in both PFS and OS compared to chemotherapy alone. Similarly, the GLOW trial, which used a CAPOX-based regimen and was conducted primarily in Asian populations, confirmed the benefit of adding zolbetuximab, reinforcing the role of CLDN18.2 as a clinically relevant therapeutic target [141].

### 3.3. Advances in Targeting KRAS for GI Tumors

With *HRAS* and *NRAS, KRAS* is part of the RAS family and it cycles between inactive (GDP-bound in this form) and active GTP-bound states, regulated by GTPase-activating proteins (GAPs, e.g., NF1) and guanine nucleotide exchange factors (GEFs, e.g., SOS1) [142,143]. When GTP-bound KRAS interacts with effectors like BRAF and CRAF, downstream MAPK signaling is triggered. Oncogenic mutations may lock KRAS protein in the GTP-bound state, driving persistent pathway activation.

At a molecular level, *KRAS* is reported to be a driver gene and it is present in approximately 20% of all cancers [144]. In particular, among human malignancies, it is particularly prevalent in pancreatic ductal adenocarcinoma and CRC, respectively representing ~90% and ~40/50% [143]. The most common hotspot mutations are pathogenic and are found in codon 12 or 13, causing p.G12D, p.G12V, and p.G12C mutations. These alterations occur in the P-loop or switch II region and may render KRAS protein constitutively active by impairing its intrinsic GTP hydrolysis.

The *KRAS* gene was previously considered undruggable, then the discovery of a cryptic pocket around cysteine 12 in *KRAS* p.G12C enabled the development of covalent inhibitors that trap KRAS in its inactive form [145]. Sotorasib and adagrasib were the first drugs developed, leading the way with impressive efficacy in NSCLC in monotherapy, though response rates have been more modest in mCRC, around 9% for sotorasib and 19% for adagrasib [146,147,148,149,150]. Mechanisms explaining this tissue specificity mirror early findings in *BRAF*-mutant CRC, where feedback EGFR activation undermined BRAF inhibitor efficacy. In *KRAS* p.G12C-mutant CRC, EGFR signaling promotes reactivation of wild-type RAS and bypasses KRAS inhibition [151]. Dual targeting of *KRAS* p.G12C and EGFR enhances outcomes in preclinical models, and clinical studies combining *KRAS* G12C inhibitors with anti-EGFR agents (cetuximab or panitumumab) have shown 2–3-fold increases in response rates compared to monotherapy [149,152,153,154,155]. In mCRC, the CodeBreaK 300 phase III trial compared sotorasib plus panitumumab versus investigator’s choice chemotherapy (regorafenib, or TAS-102) in refractory *KRAS* p.G12C-mutant mCRC, demonstrating improved progression-free survival and objective response rate with OS data still immature [156].

Beyond p.G12C, the next wave of RAS-targeted therapies addresses a broader range of *KRAS* mutations and RAS isoforms, with strategies spanning from mutation-selective inhibitors for non-G12C variants, to pan-*KRAS*-selective inhibitors targeting multiple *KRAS* mutations, and ultimately to pan-RAS inhibitors that block all RAS family members [157]. The main RAS inhibitors and their development stages are summarized in Table 2.

*KRAS* p.G12D is the most frequent KRAS alteration, accounting for 40–45% of *KRAS* mutations in pancreatic cancer and 25–45% in mCRC [158,159,160]. New drugs such as MRTX1133, small-molecule, non-covalent, and selective, designed to bind specifically to the active form of *KRAS* p.G12D, or RMC-9805, a RAS(ON) tricomplex inhibitor, have early preclinical data showing potent preclinical activity, with initial human trials now ongoing [161,162,163].

Pan-RAS inhibitors target multiple mutant RAS proteins (KRAS, NRAS, HRAS) and may also inhibit wild-type RAS. RMC-6236 exemplifies this class: it binds a conserved switch-pocket across RAS isoforms. Early-phase clinical updates at international congresses reported promising tolerability and initial tumor responses in NSCLC and pancreatic cancer, including an objective response rate of ~20% in pancreatic cancer [164,165,166].

KRAS-selective broad inhibitors target all mutant KRAS isoforms while sparing wild-type KRAS and non-KRAS RAS proteins, reducing potential toxicity. BI 2865, a newly developed non-covalent, selective inhibitor targeting the inactive state of KRAS, shows broad preclinical efficacy against diverse *KRAS* mutants, supporting the potential of pan-KRAS inhibition as a therapeutic strategy in KRAS-driven cancers [167].

### 3.4. Genetic Alteration of BRCA1/2 Genes in Pancreas

Despite advances in treatment, pancreatic cancer remains one of the most challenging malignancies. Historically, clinical trials in unselected populations have shown limited success [168]. At a genetic level, 5–7% of pancreatic cancer patients carry germline *BRCA1* or *BRCA2* mutations, which may be associated with homologous recombination deficiency (HRD). This genetic alteration can increase the sensitivity to DNA-damaging agents such as platinum-based chemotherapy [169]. In this context, the POLO trial demonstrated that maintenance therapy with olaparib in patients with *BRCA1/2* germline mutations who responded to platinum chemotherapy significantly prolonged progression-free survival, leading to approval in this setting [170,171]. However, adding PARP inhibitors to platinum-based chemotherapy upfront did not show benefit. A phase II trial comparing cisplatin–gemcitabine plus veliparib vs. chemotherapy alone was negative, indicating no added value in combining these agents [172].

Beyond germline mutations, strategies to exploit other molecular markers of HRD and novel combinations of drugs are being investigated further. For example, olaparib combined with pembrolizumab as maintenance therapy in HRD-positive patients (germline or somatic) has shown promise in early data in a recent phase II trial [173].

### 3.5. Fibroblast Growth Factor Receptor (FGFR) Alterations

Another relevant genetic target is the FGFR family that comprises four transmembrane receptor TKs that mediate essential physiological functions such as cellular proliferation, differentiation, migration, survival, and angiogenesis. FGFRs activate downstream pathways, including RAS-MAPK and PI3K-AKT, contributing to tumorigenesis when dysregulated. *FGFR* genetic alterations, such as fusions, amplifications, and point mutations, can lead to constitutive activation of these pathways and cancer progression [174,175].

In intrahepatic cholangiocarcinoma, *FGFR2* fusions are identified in approximately 10–16% of cases, representing the most frequent and therapeutically relevant *FGFR* aberration in GI cancers [176]. In another setting, gastric cancer, *FGFR2* amplification occurs in 2–9%, more commonly in diffuse-type histology and younger patients [177]. In mCRC, *FGFR1–FGFR3* alterations, including amplifications and rearrangements, are found in <5%, often in right-sided tumors without *KRAS* or *BRAF* mutations [178,179]. In pancreatic cancer, *FGFR* alterations are rare, found in fewer than 1% of cases [168]. Occasional *FGFR* aberrations have been described in hepatocellular carcinoma and esophageal cancer, although their therapeutic relevance is still unclear [180].

Several FGFR inhibitors have demonstrated clinical benefit, particularly in cholangiocarcinoma. Pemigatinib was the first FGFR inhibitor approved for previously treated FGFR2 fusion-positive intrahepatic cholangiocarcinoma, based on the FIGHT-202 trial, which showed an ORR of 35.5% [181]. Infigratinib received accelerated FDA approval based on a phase II trial showing an ORR of 23.1% but was withdrawn in 2024 pending further clinical data [182]. Futibatinib, a potent irreversible pan-FGFR1-4 inhibitor, demonstrated durable activity in the FOENIX-CCA2 study, with an ORR of 41.7%, leading to FDA approval for FGFR2-fusion-positive cholangiocarcinoma [176]. Although not approved for GI tumors, erdafitinib is FDA-approved for FGFR-altered urothelial carcinoma and is under investigation in other malignancies with FGFR aberrations [183]. Recent tumor-agnostic trials, including NCI-MATCH and RAGNAR, have evaluated erdafitinib in patients with various FGFR alterations across multiple cancer types (excluding urothelial cancer patients). In the RAGNAR trial erdafitinib demonstrated an ORR of 30.8% and a disease control rate of 73.8% across 16 solid tumor types with FGFR alterations, confirming its tumor-agnostic activity in a heavily pretreated population [184]. In the NCI-MATCH trial, despite central confirmation of *FGFR* amplifications, no objective responses were observed, with only 5 out of 18 patients (28%) included in the prespecified primary efficacy analysis, achieving stable disease as best response [185]. Those differences found between the trials, despite both investigating erdafitinib in *FGFR*-altered tumors, can be explained by the RAGNAR trial focusing on activating *FGFR* mutations and fusions, which are more likely to confer oncogene addiction, while NCI-MATCH K1 included only *FGFR* amplifications, which may not drive tumor growth and thus showed minimal clinical activity [186].

### 3.6. Other Clinically Actionable Genes

In conjunction with *FGFR2* fusions, Isocitrate dehydrogenase 1 (*IDH1)* mutations, the *BRAF* p.V600E hotspot, and others are found in approximately 30–40% of patients with cholangiocarcinoma, particularly intrahepatic subtypes [187]. Current guidelines (August 2025) recommend performing molecular testing at diagnosis or during first-line therapy to identify targeted treatment options as early as possible, especially given the availability of approved therapies in molecularly selected populations [188].

Numerous *IDH1* mutations were well-characterized in cholangiocarcinoma, especially in the intrahepatic type, occurring in approximately 13–15% of cases [189]. Authors proposed that *IDH1* mutations may lead to the production of oncometabolite 2-hydroxyglutarate, which promotes tumorigenesis (through epigenetic dysregulation). They are typically mutually exclusive with *FGFR2* fusions, another key molecular driver in intrahepatic cholangiocarcinoma [190,191]. From a therapeutic standpoint, the IDH1 inhibitor ivosidenib has demonstrated clinical efficacy in this setting: in the ClarIDHy phase III trial, ivosidenib significantly improved PFS versus placebo in patients with previously treated *IDH1* mutation. Based on this evidence, ivosidenib has been approved as a targeted therapy option for this molecular subgroup [192].

GISTs are the most common mesenchymal tumors of the gastrointestinal tract and are primarily driven by activating mutations in the *KIT* (75–80%) or *PDGFRA* (5–10%) genes [193]. Also in this case, these mutations lead to constitutive activation of the TK signaling pathway. GIST management was revolutionized with the advent of TK inhibitors: imatinib remains the standard first-line treatment for most *KIT* exon 9 mutant GISTs, while sunitinib and regorafenib are approved for second- and third-line settings, respectively.

In patients with hotspot p.D842V mutations in the *PDGFRA* gene, which are resistant to imatinib, the selective inhibitor avapritinib has shown high response rates and it is now approved for this molecular subset [194]. Additionally, newer agents such as ripretinib and bezuclastinib are being explored for resistant or refractory cases [195,196]. Molecular profiling of tumor tissue is critical at diagnosis, but ctDNA is emerging as a non-invasive tool to monitor resistance mutations and disease evolution, offering potential for real-time therapeutic adaptation [197].

The last genetic biomarkers of our review are the 5-methylthioadenosine phosphorylase (*MTAP*) deletions which are present in 10–15% of all cancers, co-occurring in 80–90% of CDKN2A-deleted malignancies [198]. It is known that *MTAP* deficiency leads to synthetic lethality when targeting PRMT5 (such as drugs such as CA240-007), which acts within the methionine metabolism pathway [199,200,201]. A phase I clinical trial with CA240-007 showed that, in heavily pretreated pancreatic cancer patients, the ORR was around 25% with the higher doses of the drug [202].

## 4. Clinical Tumor Molecular Boards in the Interpretation of Genomic Alterations

The integration of genomic information from diagnosis is essential to guide therapeutic decisions from the early diagnostics, as it can influence treatment strategies from the very beginning (i.e., neoadjuvant therapy in cases of localized disease, first-line treatment for metastatic cancers) and should be incorporated into clinical and molecular tumor boards (MTBs) in which all specialists deciding the treatment route of patients participate. The interpretation of those alterations must be made within the individual context of each patient, not only from the socio-economical point of view but also considering differences in resources and regulatory approvals across countries.

Molecular pathology reports should be clear, standardized, and clinically useful: including patient and sample details, methodology, and results with appropriate nomenclature (p. and c. for SNVs/indels, genomic coordinates, allele fraction, both partners in fusions, CNVs in tabular form). Reports should specify all analyses performed, including negative results (i.e., by providing the full list of analyzed genes for each alteration type), provide sequencing coverage limits, and flag potential hereditary variants for germline confirmation. Benign or likely benign variants should not be reported. Treatment target options and recruiting active clinical trials may be suggested in the final report [203].

MTBs are indeed central in the era of precision oncology, integrating genomic and clinical data to provide patient-tailored recommendations [204]. MTBs bring together a wide range of experts, such as molecular pathologists, genetic counselors, pharmacists, oncologists, surgeons, radiation oncologists, bioinformaticians, biostatisticians, epidemiologists, and translational scientists, among others. Each genetic and molecular alteration, such as codon-specific mutations in well-known hotspots of driver genes should be evaluated individually by MTBs, as therapeutic implications and regulatory eligibility can differ substantially across variants. Resources such as ONCOKB provide variant-level evidence that can support these discussions and guide personalized treatment decisions.

Advances in telecommunication technologies now allow virtual MTBs, enabling smaller centers to access large-scale clinical trial expertise from high-volume institutions. Each case is reviewed one by one, considering not only genomic data but also each patient’s comorbidities, previous toxicities, disease course, social context, therapeutic goals, and preferences. This collaborative effort allows the MTB to suggest the most appropriate therapeutic strategy (in a clinical trial or standard care) tailored to the patient’s tumor genomic profile but taking into account all the context of this multidimensional information. However, even when the best possible recommendation is given, important challenges persist for its application: access to therapies and clinical trials remains unequal, regulatory approval of new drugs can take time, and both treatments and genomic testing have high economical costs, contributing to global disparities across diverse regions.

## 5. Discussion and Conclusions

In the field of precision oncology and cancer drug development, GI tumors have historically played a protagonist role. Since the introduction of imatinib in the treatment of GIST patients as one of the pioneering personalized therapies in GI tumors, the constant evolution of genomic landscape knowledge and tailored treatments has expanded dramatically [92,205]. GIST, a type of tumor where precise alterations and specialized treatments have revolutionized therapy for individual patients, serves as an example of how understanding genomic background is crucial, even in less common types of cancer. Furthermore, the use of HER2 inhibitors for ERBB2-amplified gastric cancers, as well as BRAF inhibitors beyond melanoma, illustrates that specific biomarkers can have applications across different tumor types, independently of tumor site, affecting the design of clinical trials. Currently, the design shows a transition from a focus on histological criteria to the conduction of biomarker-centered basket trials (as anticipated in the Introduction section of this review). One of the first basket trials was in the context of *BRAF* p.V600 mutation, led by the VE Basket study, testing vemurafenib efficacy across diverse histologies [92]. However, this trial also highlighted that, in some scenarios, “tissue is still the issue”, as evidenced in mCRC where it was essential to combine *BRAF* p.V600-targeted therapy with EGFR inhibition [94]. Furthermore, a similar situation, evoking a sense of déjà vu, occurred with the development of *KRAS* p.G12C inhibitors for this tumor type [155]. The ongoing growth of knowledge makes it essential to find an equilibrium between tumor-agnostic biomarkers and tumor-specific biological factors as a crucial aspect in the field of cancer research. Across GI malignancies, ostensibly distinct alterations (*ERBB2/EGFR* amplifications; *KRAS/NRAS/BRAF* mutations; *FGFR2* amplifications; PIK3CA/PTEN lesions) converge on a small number of signaling axes, principally MAPK and PI3K–AKT–mTOR, supporting pathway-level therapeutic strategies and rational combinations. Likewise, immunogenicity constitutes a unifying therapeutic dimension: MSI, POLE/POLD1-ultramutated, and selected TMB-high states share neoantigen-driven susceptibility to PD-(L)1/CTLA-4 blockade, though tempered by tissue context and tumor ecology, reinforcing the need to interpret pan-tumor biomarkers within GI-specific microenvironments. In parallel, drug development strategies are evolving and being adapted to regulatory considerations and expanding biological knowledge. Innovative and dynamic clinical trial designs, such as platform trials, are increasingly being applied, as exemplified by the development of *KRAS* p.G12C inhibitors using master protocols [206].

New advancements in ctDNA technology have enabled the monitoring and study not only of spatial heterogeneity in the context of cancer as a systemic disease but also temporal heterogeneity through the course of cancer development and treatment shaping. The analysis of ctDNA has also allowed the improvement of genomic profiling of some types of GI cancers where solid biopsies are a challenge, such as biliary–pancreatic cancers [207]. Moreover, this biomarker has paved the way to novel therapeutical strategies such as EGFR inhibitor rechallenge in mCRC [208]. In addition, the use of ctDNA is allowing the improvement of biological knowledge of the resistance mechanisms of GI cancers to targeted therapy, such as the study of the genomic landscape of ctDNA in patients after progression on targeted therapy, allowing not only the understanding of the biological background but also opening doors to novel combinatory treatments to overcome drug resistance [209].

In recent years, we have also observed how relevant biomarkers in the metastatic setting have progressively shifted from use in refractory disease to first-line treatment and subsequently to localized disease and even the neoadjuvant setting. A clear example of this is MSI status in CRC. Initially, clinical benefit was demonstrated in studies including patients with refractory metastatic disease; however, more recent data have shown impressive efficacy in the first-line metastatic setting, both as monotherapy and in combination regimens.

Furthermore, emerging evidence in rectal and colon cancer suggests promising activity of these approaches even in earlier stages. Particularly in MSI rectal cancer, the introduction of immunotherapy in the neoadjuvant setting has opened the door to organ-preservation strategies, which can have a major positive impact on patients’ quality of life [33,34]. Another illustrative example is the case of *BRAF* p.V600 mCRC. Recent results from the BREAKWATER trial have shown that the combination of chemotherapy with targeted therapy in the first-line setting can double survival outcomes for this poor-prognosis subgroup [100]. It remains to be seen whether similar advances will occur in the future for other genomic biomarkers.

The incorporation of all this knowledge into clinical trial design in a dynamic way to improve translational research has allowed the development of novel platforms such as master observational protocols [210]. Moreover, the growing volume of data from both biological and clinical sources, coupled with advancements in sequencing platforms and the introduction of artificial intelligence, offers new possibilities for enhancing data analysis, biomarker discovery and the integration of complex and massive information, such as multiomic analysis [211]. Addressing these challenges to improve the knowledge of cancer biology to finally improve cancer treatment necessitates multidisciplinary teams. Collaboration plays a role in enhancing the better design of clinical trials and selecting biomarkers and treatment strategies, in a time- and cost-efficient way, to ultimately enhance survival and quality of life for patients.

In line with the feedback received during peer review, we also emphasize that the clinical implementation of genomic biomarkers critically depends on their alignment with international guidelines. Although a comprehensive comparison across all alterations would exceed the scope of this review and quickly become outdated, several key biomarkers already illustrate this process. MSI/TMB/POLE status is uniformly endorsed as predictive of response to immune checkpoint inhibitors, representing a paradigm for tumor-agnostic biomarkers. *ERBB2* amplification in gastric cancer is guideline-recognized for trastuzumab-based therapy, whereas *BRAF* V600E mutations in colorectal cancer are included with the requirement of *EGFR* co-inhibition, reflecting tissue-specific considerations. *NTRK* fusions exemplify successful tumor-agnostic approvals, while *FGFR2* amplification in gastric and biliary cancers represents an emerging biomarker with evolving guideline support. These examples highlight how guideline endorsement distinguishes established from investigational biomarkers and reinforce the dynamic interplay between discovery, clinical evidence, and translation into practice. In this manuscript, we have discussed all biomarkers present in ESMO guidelines for each tumor.

In summary, the increasing accessibility of genomic and multiomic information in GI cancers coupled with advancements in the structure of trials and the widening array of targeted treatments is influencing the direction of oncology towards a more individualized and biology-focused future resulting in improved clinical effectiveness for GI cancer patients. For this reason, we decided to provide an updated overview of actionable genomic alterations in GI cancers and discuss their implications for clinical decision making in this manuscript.

## Figures and Tables

**Figure 1 genes-16-01130-f001:**
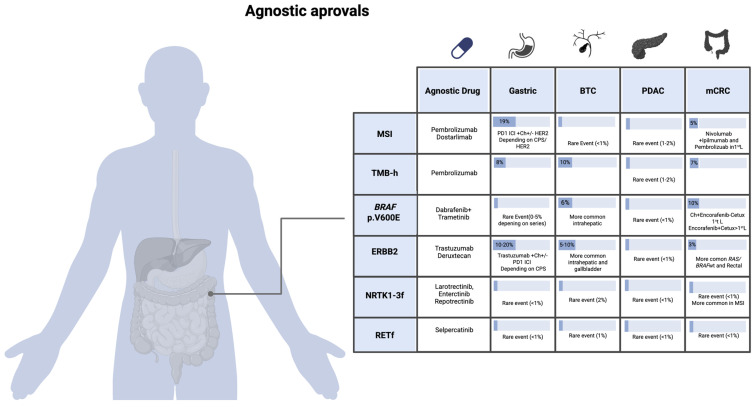
Tumor-agnostic drug approvals in GI cancers. The percentage of alterations may vary across studies. BTC: biliary tract cancer; PDAC: pancreatic ductal adenocarcinoma; mCRC: metastatic colorectal cancer; RETf: *RET* gene fusions; NTRK1-3f: *NTRK1*, *NTRK2*, *NTR3* gene fusions; MSI: microsatellite instable tumor; TMB-h: tumor mutational burden high.

**Figure 2 genes-16-01130-f002:**
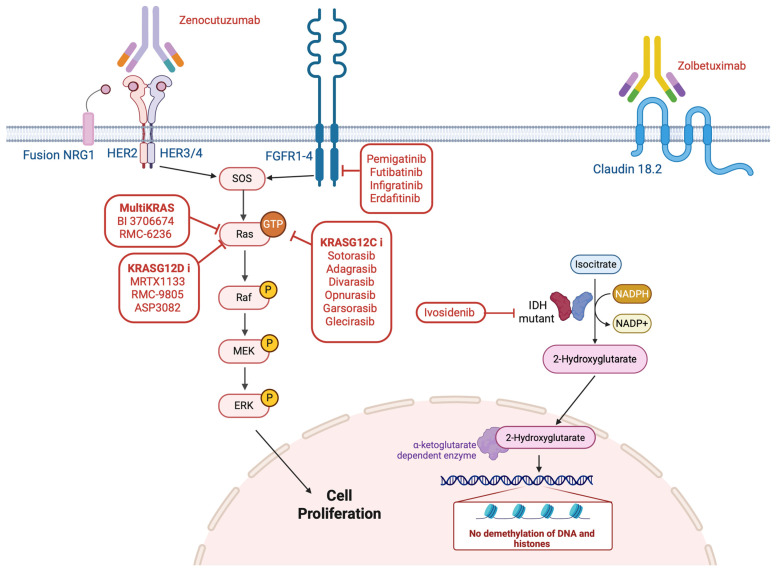
Key Non-Tumor-Agnostic Biomarkers and Their Matched Targeted Therapies in GI Cancers.

**Table 1 genes-16-01130-t001:** Summary of the efficacy of ICI in monotherapy in GI cancer pivotal trials.

Study	Study Design	Nº of Patients	MSI-H/dMMR Testing Method	ICI Regimen	Clinical Outcomes
KN-016	Phase II, mCRC and non-mCRC MSI/dMMR patients	86	Local PCR or IHC	Pembrolizumab 10 mg/kg every 2 weeks	General: ORR: 50%/CR: 21%/PR: 28/86 (33%) Colon *n* = 40; ORR 53% (5 CR, 16 PR) Ampullary *n* = 4; ORR 25% (1 CR, 1 SD, 1 PD) Cholangiocarcinoma *n* = 4; ORR 25% (1 CR, 3 SD) Gastroesophageal *n* = 5; ORR 60% (3 CR, 2 PD) Pancreas *n* = 8; ORR 63% (2 CR, 3 PR, 1 SD) Small intestine: 5; ORR 80% (2 CR, 2 PR, 1 PD)
KN-164	Phase II, refractory MSI/dMMR CRC	124	Local PCR or IHC	Pembrolizumab 200 mg every 3 weeks	ORR: 41/124 (33%) All CRC
KN-158	Phase II, non mCRC dMMR	233	Local PCR/IHC or central PCR	Pembrolizumab 200 mg every 3 weeks	General: ORR: 34% CR: 10 /PR: 24%/ SD: 18% Gastric *n* = 24; ORR 46% (4 CR, 7 PR) Cholangiocarcinoma *n* = 22; ORR 41% (2 CR, 7 PR) Pancreatic *n* = 22; ORR 18% (1 CR, 3 PR) Small intestine *n* = 19; ORR 26% (3 CR, 2 PR) Anal *n* = 1
GARNET	Phase I	327			General: ORR: 39% Colon *n* = 115; ORR = 43,5% Small intestine *n* = 23; ORR = 39% Pancreatic *n* = 12; ORR:41% Gastric *n* = 22 ORR 45%

ORR: Overall Response Rate; CR: Complete Response; PR: Partial Response; PD: Progressive Disease; SD: Stable Disease; IHC: Immunohistochemistry; PCR: Polymerase Chain Reaction.

**Table 2 genes-16-01130-t002:** Summary of main RAS inhibitors.

Drug Name	Target	Development Stage
Sotorasib (AMG510 Amgen)	KRAS p.G12C (OFF)	Approved
Adagrasib (MRTX849 Mirati)	KRAS p.G12C (OFF)	Approved
Divarasib (GDC-6036 Genentech/Roche)	KRAS p.G12C (OFF)	Phase III
Opnurasib (JDQ443 Novartis)	KRAS p.G12C (OFF)	Phase III
Garsorasib (D-1553 InventisBio)	KRAS p.G12C (OFF)	Phase II
Glecirasib (JAB-21822 Jacobio)	KRAS p.G12C (OFF)	Phase II
GFH925 (GenFleet)	KRAS p.G12C (OFF)	Phase II
YL-15293 (Shanghai Yingli)	KRAS p.G12C (OFF)	Phase II
HS-10370 (Jiangsu Hansoh)	KRAS p.G12C (OFF)	Phase II
LY3537982 (Lilly)	KRAS p.G12C (OFF)	Phase II
BI 1823911 (Boehringer Ingelheim)	KRAS p.G12C (OFF)	Phase II
BPI0421286 (Belta)	KRAS p.G12C (OFF)	Phase II
GH35 (Suzhou Genhouse Bio)	KRAS p.G12C (OFF)	Phase II
GEC255 (GenEros Biopharma)	KRAS p.G12C (OFF)	Phase II
MK-1084 (Merck)	KRAS p.G12C (OFF)	Phase II
D3S-001 (D3 Bio)	KRAS p.G12C (OFF)	Phase II
HBI-2438 (HuyaBio)	KRAS p.G12C (OFF)	Phase II
SY-5933 (Shouyao Holdings)	KRAS p.G12C (OFF)	Phase II
JNJ-74699157 (Janssen)	KRAS p.G12C (OFF)	Discontinued
RMC-6291 (Revolution Medicines)	KRAS p.G12C (ON)	Phase I
MRTX1133 (Mirati)	KRAS p.G12D	Phase I
RMC-9805 (Revolution Medicines)	KRAS p.G12D	Phase I
ASP3082 (Astellas)	KRAS p.G12D	Phase I
HRS-4642 (Jiangsu Hengrui)	KRAS p.G12D	Phase I
INCB161731 (Incyte)	KRAS p.G12D	Phase I
BI 3706674 (Boehringer Ingelheim)	Multi-KRAS	Phase I
RMC-6236 (Revolution Medicines)	Multi-KRAS	Phase I

## Data Availability

No new data were created or analyzed in this study. Data sharing is not applicable to this article.

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
