# Peer review of "Clinical Actionability of Genes in Gastrointestinal Tumors"

_genes, 2025, doi:10.3390/genes16101130_

Round 1
Reviewer 1 Report
Comments and Suggestions for Authors
For participants in tumor boards, the question of identifying actionable gene variants outside of mandatory companion diagnostics is particularly important. It would therefore be advisable to include a chapter on driver mutations that are highly likely to be actionable outside the existing approval status with available drugs, e.g., in the case of BRAF, in addition to p.V600E, also p V600K p.V600D, p.V600R, and p. V600M, or in the case of FGFR2 p. C382R. Such an overview would significantly enhance the article's usefulness in molecular tumor boards.
Furthermore, a brief overview of the methodology and assessment of the quality of genomic profiling would also be helpful for clinicians/participants in molecular tumor boards in order to be able to classify NGS reports accordingly. Another example is the question of when DNA-based sequencing and when RNA-based sequencing is mandatory.
Author Response
We would like to express our profound gratitude for the Reviewer’s constructive feedback on the revision. A very valuable point was raised regarding tumor boards. We agree that the interpretation and integration of genomic information in clinical and molecular tumour boards (MTBs) is a highly relevant topic, and there is a need to include genomic information from the very beginning (ideally from patient diagnosis) to take the best clinical decisions for patients. In response to these valuable comments, we have added a section discussing the integration of genomic data in MTBs (lines 743-781) and the role of MTBs in this regard. About the consideration of genomic alterations beyond guideline-approved variants (in the European context), this issue is of particular relevance for pre-screening programs in centers that have the capacity to enroll patients in clinical trials. Therefore, an additional clarification has been provided regarding this critical aspect of daily clinical decision-making.
About the consideration of genomic alterations beyond guideline-approved variants (in the European context), such as BRAF example, this issue is of particular relevance for pre-screening programs in centers that have the capacity to enroll patients in clinical trials. Therefore, an additional clarification has been provided regarding this critical aspect of daily clinical decision-making (line 765-769). We hope these revisions satisfactorily address the Reviewers’ comments
Reviewer 2 Report
Comments and Suggestions for Authors
The manuscript by Saoudi et al. provides an extensive and well-structured review of actionable genomic alterations in gastrointestinal (GI) cancers. It covers both tumor-agnostic and tumor-specific biomarkers, emphasizing their therapeutic implications, clinical trial evidence, and emerging strategies. The review is comprehensive, timely, and aligns well with the growing need to integrate molecular oncology into routine GI cancer management. While the manuscript has significant merit and is potentially publishable, important improvements are required to enhance scientific depth, clarity, and clinical applicability. The breadth of information is impressive, but the review would benefit from deeper mechanistic insights, critical appraisal of controversies, and integration of recent high-impact studies.
Major comments:
- The review effectively catalogs actionable alterations but often lacks mechanistic depth. For example, in discussing MSI-H gastric cancers, the authors could better explore the interaction between epigenetic silencing (MLH1 methylation), immune microenvironment dynamics, and therapeutic resistance. Similarly, in KRAS and BRAF sections, further explanation of pathway reactivation mechanisms would contextualize combination strategies more clearly.
- The review tends to be descriptive rather than evaluative. For instance, while summarizing clinical trial outcomes, the authors should more explicitly compare the robustness of evidence across biomarkers (e.g., well-validated HER2 vs. emerging FGFR2 amplifications). Highlighting unresolved controversies—such as the optimal cutoff for tumor mutational burden (TMB) or the uncertain utility of NRG1 testing—would strengthen the review.
- Gastric cancer is discussed mainly in terms of single biomarkers. However, growing evidence indicates that histological and anatomical location (proximal vs. distal tumors; intestinal vs. diffuse histotypes) critically shape transcriptional programs, mutational landscapes, and therapeutic vulnerabilities. This dimension is currently underdeveloped in the review.
- The manuscript does not address gene regulatory networks and master regulators, which have recently been shown to define histotype-specific and location-specific biology in gastric cancer. This omission leaves out a layer of biological complexity that has direct translational implications for biomarker-guided therapy.
- At times, the manuscript reads as an accumulation of separate biomarker reviews. A more cohesive narrative that synthesizes cross-cutting themes (e.g., converging pathways such as MAPK or PI3K across multiple alterations; immunogenicity as a unifying therapeutic axis) would improve readability and impact.
- The review should more explicitly connect findings to current NCCN, ESMO, and ASCO guidelines. This would help clinicians translate genomic findings into practice and highlight where evidence is strong enough to warrant guideline inclusion versus where it remains investigational.
- Current figures summarize therapeutic approvals, but the review could benefit from a schematic integrating tumor-agnostic and tumor-specific biomarkers into a single precision oncology workflow for GI tumors.
- To strengthen the discussion of gastric cancer biology, I recommend the authors integrate and cite the following recent works: doi:10.3390/cancers14194961 - This study demonstrates how gene regulatory networks and master regulators distinguish intestinal vs. diffuse gastric cancer, with implications for personalized therapy. It should be discussed in relation to histotype-specific actionability. doi:10.3390/curroncol32080424 - This paper highlights location-specific transcriptional programs in gastric cancer and identifies master regulators with prognostic and therapeutic relevance. It directly supports the need for stratified genomic approaches beyond single biomarkers.
Minor Comments:
- Add a short “Future Directions” section focusing on how multi-omic integration, artificial intelligence, and regulatory network analysis could enhance biomarker discovery in GI cancers.
- Discuss challenges in real-world implementation (cost, access, testing heterogeneity).
- Include equity considerations, especially regarding access to biomarker testing and targeted therapies in low- and middle-income countries.
- Ensure adherence to EQUATOR guidelines relevant for reviews
Author Response
The manuscript by Saoudi et al. provides an extensive and well-structured review of actionable genomic alterations in gastrointestinal (GI) cancers. It covers both tumor-agnostic and tumor-specific biomarkers, emphasizing their therapeutic implications, clinical trial evidence, and emerging strategies. The review is comprehensive, timely, and aligns well with the growing need to integrate molecular oncology into routine GI cancer management. While the manuscript has significant merit and is potentially publishable, important improvements are required to enhance scientific depth, clarity, and clinical applicability. The breadth of information is impressive, but the review would benefit from deeper mechanistic insights, critical appraisal of controversies, and integration of recent high-impact studies.
We sincerely thank the Reviewer for the kind words and for recognizing both the value of our work and the considerable effort invested in critically handling and organizing such a large body of information. As also reported in this overview of our work, our primary goal was indeed to provide readers with a clear, immediately accessible overview of actionable genomic alterations in gastrointestinal (GI) cancers.
We fully appreciate the importance of incorporating deeper mechanistic insights. On the other hand, given the breadth of genes and alterations already covered, adding this level of detail would substantially expand the manuscript and risk diluting its intended purpose that is, at least as originally conceived, to serve as a practical, concise, and clinically oriented reference. In our view, a detailed mechanistic discussion might be better suited to focused reviews on specific pathways or gene families. We can evaluate this work in the future.
Regarding the relevant controversies, we have made every effort to highlight key scientific discussions and to integrate critical commentary where appropriate, ensuring that the manuscript maintains both scientific rigor and clinical applicability while remaining accessible to the intended readership.
Major comments:
- The review effectively catalogs actionable alterations but often lacks mechanistic depth. For example, in discussing MSI-H gastric cancers, the authors could better explore the interaction between epigenetic silencing (MLH1 methylation), immune microenvironment dynamics, and therapeutic resistance. Similarly, in KRAS and BRAF sections, further explanation of pathway reactivation mechanisms would contextualize combination strategies more clearly.
We sincerely appreciated the Reviewer for this valuable suggestion. Mechanistic insights are indeed addressed within the manuscript, although with an overview intention rather that with in-depth details (please check for instance the initial part of section 2.1, the second paragraph of section 2.3). Furthermore, the mechanistic activations of BRAF were addressed in the 5th paragraph from the same section by dissecting the tissue specificity of BRAF monotherapy inhibition (other details in the first paragraph of section 3.2). As previously indicated, in our original view, the manuscript was conceived to provide sufficient biological context to explain clinical responses, without extending into mechanistic detail beyond the scope of a clinical review (we are evaluating these aspects for a future work).
- The review tends to be descriptive rather than evaluative. For instance, while summarizing clinical trial outcomes, the authors should more explicitly compare the robustness of evidence across biomarkers (e.g., well-validated HER2 vs. emerging FGFR2 amplifications). Highlighting unresolved controversies—such as the optimal cutoff for tumor mutational burden (TMB) or the uncertain utility of NRG1 testing—would strengthen the review.
We thank the Reviewer for this constructive comment and for encouraging us to strengthen the evaluative component of our review. We fully agree that highlighting controversies and differences in the robustness of evidence across biomarkers is crucial to provide readers with a balanced perspective.
As an example, we dedicated Section 2.2. TMB-High (Tumor Mutational Burden ≥10 mut/Mb) precisely to this issue. In this paragraph, we summarize how TMB has emerged as a potential predictive biomarker for immune checkpoint inhibitors (ICIs), while also acknowledging the challenges and ongoing debates. Specifically, we describe its biological rationale (increased neoantigen load and T cell activation), the methodological differences between whole-exome sequencing and targeted panels, and the variability across GI cancer subtypes, where TMB-H prevalence remains generally lower than in melanoma or lung cancer.
We also report data from the pivotal KEYNOTE-158 trial, which underpinned the FDA approval of pembrolizumab for TMB-H tumors (≥10 mut/Mb), and we critically comment on the modest benefit observed in GI cancers (e.g., only one responder among 14 anal cancer cases). Importantly, we discuss how a subset of TMB-H colorectal cancers lacking MSI-H or POLE/POLD1 mutations shows limited response to ICIs, thereby underscoring the limitations of using TMB as a pan-tumor biomarker. We further highlight the unresolved issue of the optimal cutoff (≥10 vs. 13–20 mut/Mb) and the differences between blood-derived TMB and tissue-based TMB, which add complexity to its clinical interpretation.
In our view, this detailed analysis of TMB exemplifies our effort to go beyond a descriptive summary, by explicitly evaluating the strength of evidence, highlighting unresolved controversies, and contextualizing the clinical utility of this biomarker within the heterogeneous landscape of GI cancers.
- Gastric cancer is discussed mainly in terms of single biomarkers. However, growing evidence indicates that histological and anatomical location (proximal vs. distal tumors; intestinal vs. diffuse histotypes) critically shape transcriptional programs, mutational landscapes, and therapeutic vulnerabilities. This dimension is currently underdeveloped in the review.
We sincerely thank the Reviewer for this insightful comment, which we believe has significantly improved our manuscript. In response, we have added a paragraph on the molecular classification of gastric cancer in Section 2.1.2.(lines 228-239) to address histological and anatomical factors influencing tumor biology and therapeutic implications. Specifically, we now describe the four molecular gastric/gastro-esophageal cancer subtypes defined by the TCGA classification (EBV-positive, MSI, genomically stable, and chromosomal instability tumors), each associated with distinct genetic and epigenetic alterations. Furthermore, we highlight recent studies underscoring the importance of gene regulatory networks and master regulators in distinguishing intestinal versus diffuse histotypes, as well as location-specific transcriptional programs with prognostic and therapeutic implications. We are grateful for the Reviewer’s suggestion, as it allowed us to move beyond a single-biomarker perspective and to provide a more nuanced overview that reflects the complexity and clinical relevance of gastric cancer biology.
– The manuscript does not address gene regulatory networks and master regulators, which have recently been shown to define histotype-specific and location-specific biology in gastric cancer. This omission leaves out a layer of biological complexity that has direct translational implications for biomarker-guided therapy.
We thank the Reviewer for this thoughtful suggestion. While we agree that gene regulatory networks and master regulators provide important biological insights, they remain relatively distant from current clinical practice. Given the clinical focus of our review and that we are evaluating these aspects for a future work, we respectfully preferred not to expand on this aspect, though we acknowledge its growing relevance for future translational work.
- At times, the manuscript reads as an accumulation of separate biomarker reviews. A more cohesive narrative that synthesizes cross-cutting themes (e.g., converging pathways such as MAPK or PI3K across multiple alterations; immunogenicity as a unifying therapeutic axis) would improve readability and impact.
We thank the Reviewer for highlighting the value of a more cohesive, cross-cutting narrative. We agree with this point and, while retaining the section-by-section framework to preserve navigability for clinicians and translational researchers, we have strengthened the integrative synthesis in the Discussion and Conclusions (lines 803-811). These sections now explicitly emphasize (i) pathway-level convergence across alterations (e.g., MAPK and PI3K–AKT–mTOR) and its therapeutic implications, and (ii) immunogenicity as a unifying therapeutic axis (MSI, POLE/POLD1, TMB-high) modulated by tissue context. Importantly, this comment also helped us to better highlight the rationale underlying the ongoing transition from histology-based to basket-type clinical trials, where molecular mechanisms and biological convergence rather than tumor site are increasingly guiding trial design. We are grateful to the Reviewer for this suggestion, which has further reinforced the coherence and translational relevance of our review.
- The review should more explicitly connect findings to current NCCN, ESMO, and ASCO guidelines. This would help clinicians translate genomic findings into practice and highlight where evidence is strong enough to warrant guideline inclusion versus where it remains investigational.
We thank the Reviewer for this suggestion, which has strengthened the clinical perspective of our manuscript. As recommended, we have added a dedicated passage in the Discussion (lines 854–868) emphasizing the role of guideline alignment as a key element for clinical translation. In this section, we highlight several pivotal biomarkers, MSI/TMB/POLE, ERBB2, BRAF V600E, NTRK, and FGFR2, as examples where international guidelines (NCCN, ESMO, ASCO) already influence therapeutic decision-making or are under evolving consideration. Furthermore, we implemented a sentence about the biomarkers considered in the ESMO guidelines and recommendation in the Introduction section (lines 77-80). We believe these additions further improve the translational value and clarity of the review.
- Current figures summarize therapeutic approvals, but the review could benefit from a schematic integrating tumor-agnostic and tumor-specific biomarkers into a single precision oncology workflow for GI tumors.
We sincerely thank the Reviewer for this constructive idea. We tried to generate a unique Figure for both tumor-agnostic and tumor-specific biomarkers, but the figure became overcrowded and less informative. For this reason, we respectfully preferred to keep Figure 2 covering the main tumor-specific biomarkers and Figure 1 those of tumor-agnostic markers. Moreover, the manuscript itself addresses these two categories in detail in separate sections (pages 2-12 for tumor-agnostic and pages 12-18 for tumor-specific biomarkers), so that the reader can easily appreciate both perspectives. At present, we have not identified an alternative format that would enhance clarity and utility for the reader beyond the current approach.
- To strengthen the discussion of gastric cancer biology, I recommend the authors integrate and cite the following recent works: doi:10.3390/cancers14194961 - This study demonstrates how gene regulatory networks and master regulators distinguish intestinal vs. diffuse gastric cancer, with implications for personalized therapy. It should be discussed in relation to histotype-specific actionability. doi:10.3390/curroncol32080424 - This paper highlights location-specific transcriptional programs in gastric cancer and identifies master regulators with prognostic and therapeutic relevance. It directly supports the need for stratified genomic approaches beyond single biomarkers.
We thank the Reviewer for suggesting these relevant papers. We have now added this information and references in the paraph about molecular classification of gastric cancer: references 64 and 65(lines 235-239).
64 Russi S, Marano L, Laurino S, Calice G, Scala D, Marino G, et al. Gene Regulatory Network Characterization of Gastric Cancer’s Histological Subtypes: Distinctive Biological and Clinically Relevant Master Regulators. Cancers (Basel) [Internet]. 2022 Oct 1 [cited 2025 Sep 2];14(19):4961. Available from: https://www.mdpi.com/2072-6694/14/19/4961/htm and
65 Marano L, Sorrenti S, Malerba S, Skokowski J, Polom K, Girnyi S, et al. Different Master Regulators Define Proximal and Distal Gastric Cancer: Insights into Prognosis and Opportunities for Targeted Therapy. Current Oncology 2025, Vol 32, Page 424 [Internet]. 2025 Jul 28 [cited 2025 Sep 2];32(8):424. Available from: https://www.mdpi.com/1718-7729/32/8/424/htm
Minor Comments:
- Add a short “Future Directions” section focusing on how multi-omic integration, artificial intelligence, and regulatory network analysis could enhance biomarker discovery in GI cancers.
We thank the Reviewer for this relevant point. A concise paraph regarding this issue is already present in the Discussion section (from the line 843 to 853).
- Discuss challenges in real-world implementation (cost, access, testing heterogeneity).
We thank the Reviewer for this important point. To address it, we have added a new Section 4 (Clinical Tumor Molecular Boards in the interpretation of genomic alterations, lines 743–782). In this section, we discuss the central role of Molecular Tumor Boards (MTBs) in translating genomic information into therapeutic decisions, and we explicitly highlight the challenges of real-world implementation. We emphasize that genomic interpretation must always be contextualized to each patient, not only from a clinical perspective but also considering socio-economic factors, resource availability, and regulatory differences across countries. We further underline that MTBs integrate a wide range of expertise (pathologists, oncologists, surgeons, genetic counselors, bioinformaticians, etc.) to provide patient-tailored recommendations, supported by resources such as OncoKB. Importantly, we also note that, despite these efforts, significant barriers remain: access to therapies and clinical trials is unequal, regulatory approval of new drugs can be slow, and both molecular testing and treatments carry substantial costs—factors that collectively contribute to disparities across regions.
We believe this addition strengthens the manuscript by explicitly addressing the Reviewer’s concern regarding cost, access, and heterogeneity in testing.
- Include equity considerations, especially regarding access to biomarker testing and targeted therapies in low- and middle-income countries.
We thank the Reviewer for raising this crucial point. Equity considerations are now explicitly discussed in the newly added Section 4 (Clinical Tumor Molecular Boards in the interpretation of genomic alterations, lines 743–781). In this section, we emphasize that genomic interpretation must be contextualized not only to the patient’s clinical profile but also to socio-economic factors, resource availability, and regulatory frameworks across different countries. We also highlight how unequal access to testing and targeted therapies, together with the high costs of drugs and diagnostics, contributes to disparities that are particularly relevant in low- and middle-income countries.
- Ensure adherence to EQUATOR guidelines relevant for reviews
The manuscript has been checked against the SANRA guidelines for narrative reviews, and all Authors confirm that it complies with the key criteria, including justification of the article’s importance, focused aims, thorough literature search with primary references, and a well-structured presentation.
once you add the ESMO references (see track changes in the introduction) I believe this becomes 65